# Quantitative measurement of diffusion-weighted imaging signal using expression-controlled aquaporin-4 cells: Comparative study of 2-compartment and diffusion kurtosis imaging models

Akiko Imaizumi[1]*, Takayuki Obata[1]*, Jeff Kershaw[1], Yasuhiko Tachibana[1], Yoichiro Abe[2], Sayaka Shibata[1], Nobuhiro Nitta[1], Ichio Aoki[1], Masato Yasui[2], Tatsuya Higashi[1]

**1** Department of Molecular Imaging and Theranostics, Institute for Quantum Medical Science, National Institutes for Quantum Science and Technology, Chiba, Chiba, Japan, **2** Department of Pharmacology, Keio University School of medicine, Shinjuku, Tokyo, Japan

* imaizumi.akiko@qst.go.jp (AI); obata.takayuki@qst.go.jp (TO)

**Data Availability Statement:** All relevant data are within the paper.

## Abstract

The purpose of this study was to compare parameter estimates for the 2-compartment and diffusion kurtosis imaging models obtained from diffusion-weighted imaging (DWI) of aquaporin-4 (AQP4) expression-controlled cells, and to look for biomarkers that indicate differences in the cell membrane water permeability. DWI was performed on AQP4-expressing and non-expressing cells and the signal was analyzed with the 2-compartment and diffusion kurtosis imaging models. For the 2-compartment model, the diffusion coefficients ($D_f$, $D_s$) and volume fractions ($F_f$, $F_s$, $F_f = 1-F_s$) of the fast and slow compartments were estimated. For the diffusion kurtosis imaging model, estimates of the diffusion kurtosis (K) and corrected diffusion coefficient (D) were obtained. For the 2-compartment model, $D_s$ and $F_s$ showed clear differences between AQP4-expressing and non-expressing cells. $F_s$ was also sensitive to cell density. There was no clear relationship with the cell type for the diffusion kurtosis imaging model parameters. Changes to cell membrane water permeability due to AQP4 expression affected DWI of cell suspensions. For the 2-compartment and diffusion kurtosis imaging models, $D_s$ was the parameter most sensitive to differences in AQP4 expression.

## Introduction

We have previously reported that diffusion-weighted imaging (DWI) signal is sensitive to the cell membrane water permeability of aquaporin-4 (AQP4) expressing and non-expressing cells [1]. Aquaporin is a membrane protein that passively facilitates the transport of water molecules between the inside and outside of the cell according to the osmotic gradient [2]. Aquaporin

**Funding:** This work was supported by grants from KAKENHI (15H04910, TO received. 20K08150, JK received.), and from the Ministry of Education, Culture, Sports, Science and Technology (MEXT), Japanese government. The funders had no role in study design, data collection and analysis, decision to publish, or preparation of the manuscript. There was no additional external funding received for this study.

**Competing interests:** The authors have declared that no competing interests exist.

channels are distributed throughout the body and maintain the distribution of water. Thirteen types of channels have been identified in mammals, with each type distributed in specific tissues and performing a particular physiological function. Aquaporin channels are known to be associated with various diseases. For example, neuromyelitis optica (NMO) is an autoimmune disease targeting AQP4 [3]. AQP4 is also associated with brain tumors [4] and neurodegenerative diseases including Alzheimer's disease [5]. It is also known that changes to the expression of AQP4 can alter the accumulation of brain edema in ischemia [6, 7]. Other aquaporin subtypes are involved in various diseases such as tumors [8], cataracts [9], and nephrogenic diabetes insipidus [10, 11]. Unfortunately, a clinical imaging method that can evaluate aquaporin expression *in vivo* has not yet been established. Moreover, as there are many subtypes of aquaporin with diverse functions, it is probably more reasonable to evaluate the cell membrane water permeability rather than the expression of aquaporin itself. It is therefore expected that an imaging technique that can quantitatively evaluate changes in cell membrane water permeability will be useful in disease diagnosis.

Various models have been proposed to analyze DWI, but the relationship between the biological tissue structure and the signal remains unclear. The most common model for the quantitative analysis of DWI is the apparent diffusion coefficient (ADC) model, where the ADC is estimated by fitting the DWI signal to a mono-exponential signal equation with respect to b-value. However, as the signal deviates from mono-exponential decay at high b-values, a bi-exponential 2-compartment signal model (2Comp) is often used as an alternative. In this model, water molecules are divided into fast and slow diffusion compartments with volume fractions ($F_f$, $F_s$) and diffusion coefficients ($D_f$, $D_s$) corresponding to each compartment [12]. The ADC model assumes that the diffusion of water molecules is Gaussian, but in biological tissues the diffusion is non-Gaussian due to restriction of molecular motion by microstructures such as the cell membrane. In that case, the diffusion kurtosis (K) parameter of the diffusion kurtosis imaging model (DKm) may be useful for characterizing the degree of deviation from Gaussian behavior [13]. K increases as the complexity of the tissue structure increases and provides information that differs from that given by the ADC [13–15]. However, because there is no clear well-established connection between the DKm model and the biological reality, the mechanism by which changes in the tissue affect the DKm parameters is uncertain.

The purpose of this study was to compare parameter estimates for the 2Comp and DKm models obtained from DWI of AQP4 expressing and non-expressing cells, and to look for biomarkers that indicate differences in cell membrane water permeability.

## Materials and methods

### Subjects

Chinese hamster ovary (CHO) cells (RCB0285, obtained from RIKEN BRC) stably transfected with either the AQP4 expression vector pIRES2-EGFP, where a unique AflII site had been modified to an EcoRI by linker ligation containing mAQP4M1 cDNA (AQP4), or the empty vector (Control) were prepared as described in a previous study [1]. The cells were centrifuged at 78.7 x g for 5 min at 4°C and suspended in phosphate buffered saline (PBS) in PCR tubes. A suspension of 0.2 ml contained $2.5 \times 10^7$ cells.

### DWI acquisition

A 7T animal MRI system (Kobelco with Bruker BioSpin, Japan) was used. Cell suspensions were placed upright in the center of the scanner. DWI was performed using a pulsed-gradient spin-echo (PGSE) sequence with multi-shot echo planar imaging (EPI) acquisition (repetition time (TR): 3000ms; echo time (TE): 90ms; matrix: 128x128; spatial resolution: 0.2x0.2 mm$^2$;

slice thickness: 2mm). The separation of the onset of the motion probing gradient (MPG) lobes ($\Delta$) and the duration of the lobes ($\delta$) were 25ms and 7ms, respectively. The b-value was increased from 0 to 8000 s/mm$^2$ in 14 steps (0, 2, 250, 500, 750, 1000, 1500, 2000, 3000, 4000, 5000, 6000, 7000, and 8000 s/mm$^2$).

A set of cell samples (AQP4 and Control) were selected for each experiment. DWI was performed on 6 sets of samples.

### DWI analysis

The data was analyzed with both the 2Comp and DKm models.

For the 2Comp model, the diffusion coefficients ($D_f$, $D_s$) and the volume fraction of the slow compartment, $F_s$, were used as the unknown parameters in the fitting procedure, and afterwards the volume fraction of the fast compartment, $F_f$, was estimated using the relationship $F_s + F_f = 1$. Nonlinear least squares was used to fit the following bi-exponential equation

$$S(b) = S_0(F_f e^{-bD_f} + F_s e^{-bD_s}) \tag{1}$$

where S(b) and $S_0$ are the signals with and without an applied MPG, respectively, and b is the b-value.

Fitting to the DKm model was performed using DW images with b-values in the range from 250 to 2000 s/mm$^2$. Outside of this range, the low b-value *in vivo* images may be affected by the intravoxel incoherent motion of blood, and high b-value images decrease the precision of the fitting [16]. The data was fitted to the following quadratic equation

$$S(b) = S_0 e^{\left(-bD + \frac{1}{6}b^2 D^2 K\right)} \tag{2}$$

and pixel-by-pixel estimates of K and the corrected diffusion coefficient (D) were obtained.

### Statistical analysis

The DWI parameter estimates obtained for each model were compared with respect to AQP4 expression.

As the ratio of the intra- to extra-cellular volumes (i.e. cell density) depends on position in the PCR tube after centrifuging, the dependence of the parameter estimates on depth was also evaluated [1]. Eight rectangular (6 pixels x 2 pixels) regions-of-interest (ROIs) were drawn on each cell sample (Fig 1), and the mean signal intensity was calculated for each ROI. The ROIs were numbered I, II, III. . .VIII from top to bottom.

Statistical analyses were performed with MATLAB version R2015a (Math Works Inc., Natick, MA). Analysis of covariance (ANCOVA) was used to compare the DWI parameter estimates between the AQP4 and Control samples using the depth as a covariate. $P < 0.05$ was considered to be statistically significant.

### Results

A good signal-to-noise ratio of about 4.44 at b = 8000 s/mm$^2$ was obtained for all of the samples. There was a clear difference in the b-value-dependent signal of the AQP4 and Control samples (Fig 2). Separate ADC maps were calculated for a low b-value range of 0–1500 s/mm$^2$ and a high b-value range of 4000–8000 s/mm$^2$ (Fig 3). There is no clear difference in the maps with respect to cell type for the low b-value range, but there is a contrast in the vertical direction that probably corresponds to cell density. On the other hand, for the high b-value range map there is no dependence on depth, but there is a clear difference between the cell types (Fig 3).

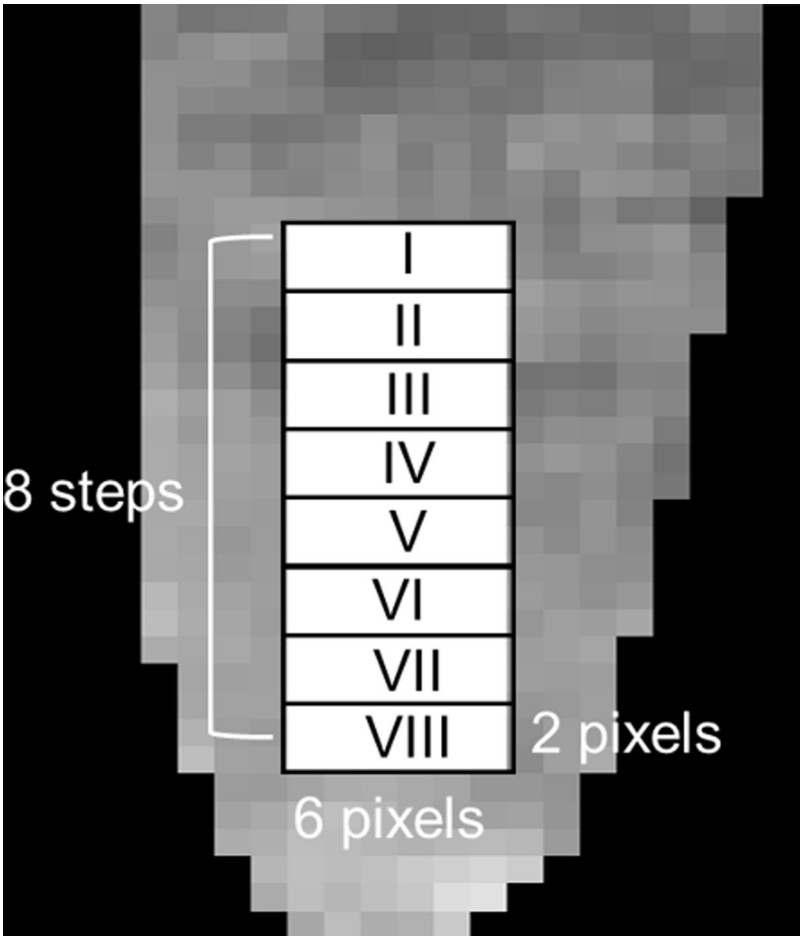

**Fig 1. Regions-of-interest (ROIs).** Eight rectangular (6 pixels x 2 pixels) ROIs were drawn on images of each cell sample, and the mean signal intensity was calculated for each ROI. The ROIs were labeled as I, II, III. . .VIII from top to bottom.

The b-value-dependent signal changes were analyzed with the 2Comp and DKm models for each of the separate ROIs (Fig 4 and Table 1).

For the 2Comp model parameters $D_s$ and $F_s$ there was a significant difference between the AQP4 and Control samples ($P<0.05$). There was also a significant dependence on depth for $F_s$ ($P<0.0001$). For the DKm model parameters K and D there was no significant difference between the AQP4 and Control samples (P = 0.232 for K and P = 0.403 for D). However, there was a significant dependence on depth for both D and K ($P<0.0001$). For K, there was also a significant interaction between the cell type and depth (P = 0.029).

## Discussion

### Summary

In this study, the 2Comp model parameters $D_s$ and $F_s$ showed a clear difference with respect to cell type. However, as $F_s$ was also affected by the depth, this suggests that $D_s$ may be a more reliable biomarker for cell-type-related differences. For the DKm model parameters, there was no significant difference in the estimates corresponding to cell type.

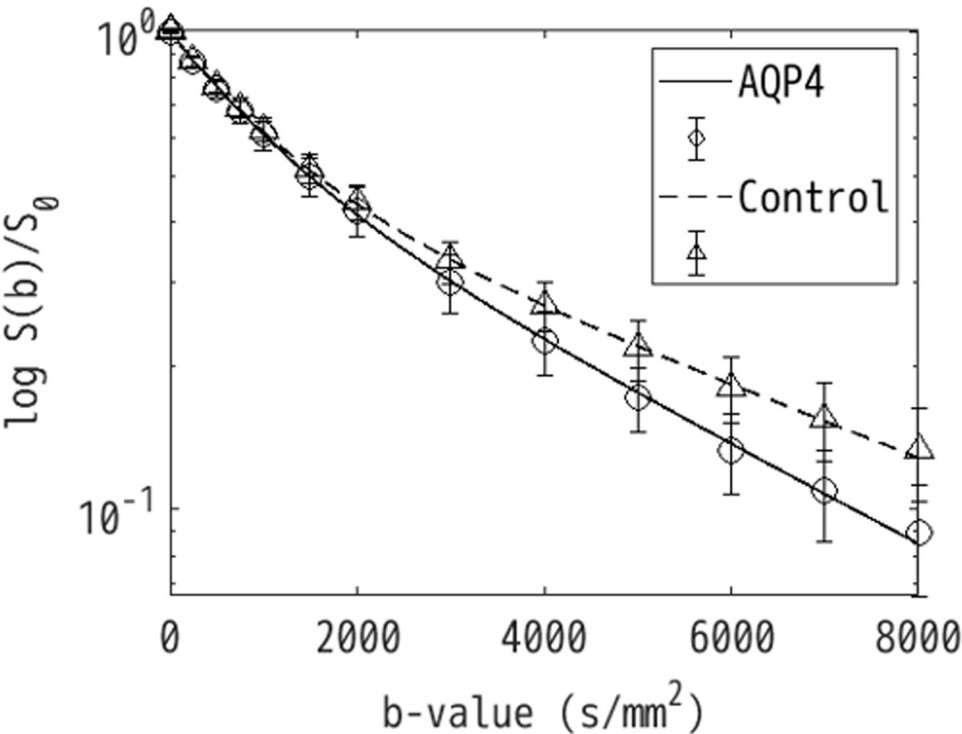

**Fig 2. Diffusion-weighted imaging (DWI) signal versus b-value for the aquaporin-4-expressing (AQP4) and -non-expressing (Control) cell samples.** The data was normalized by the b = 0 data. The solid and dashed lines indicate the AQP4 and Control data, respectively. The circles and triangles indicate the average signals across samples with the error bars corresponding to standard deviation. There is a clear difference in the decay of the AQP4 and Control data with respect to b-value.

## 2Comp model

In the framework of the 2Comp model water diffusion is divided into fast and slow compartments. The slow compartment is thought to correspond to intracellular and para-cell-membranous water molecules, and molecular diffusion is restricted by intracellular microstructures and the cell membrane [12]. It might therefore be expected that a difference in cell membrane water permeability due to the presence or absence of AQP4 expression would most affect $D_s$. It could also be anticipated that a change in the intra-/extra-cellular volume ratio as a function of depth would most affect $F_s$. The observations of this study were consistent with these expectations as the estimates of $D_s$ were significantly different for the AQP4 and Control samples (P = 0.0003, Table 1), while $F_s$ increased with depth (P<0.0001, Table 1). It is possible that $D_s$ might be an effective biomarker for quantitatively evaluating cell membrane water permeability. In contrast, even though the results suggest that $F_s$ could be useful as a means to characterize cell density after centrifuging, it should be remembered that $F_s$ also had a significant dependence on cell type (P = 0.005, Table 1). This result is difficult to explain with the 2Comp model and further studies are required.

## DKm model

The DKm model parameter K increases as the tissue structure becomes more complex. Changes in K have been reported in diseases including tumors, cerebral ischemia, and neurodegenerative diseases [16–18]. In this study, no clear dependence on cell type was detected for

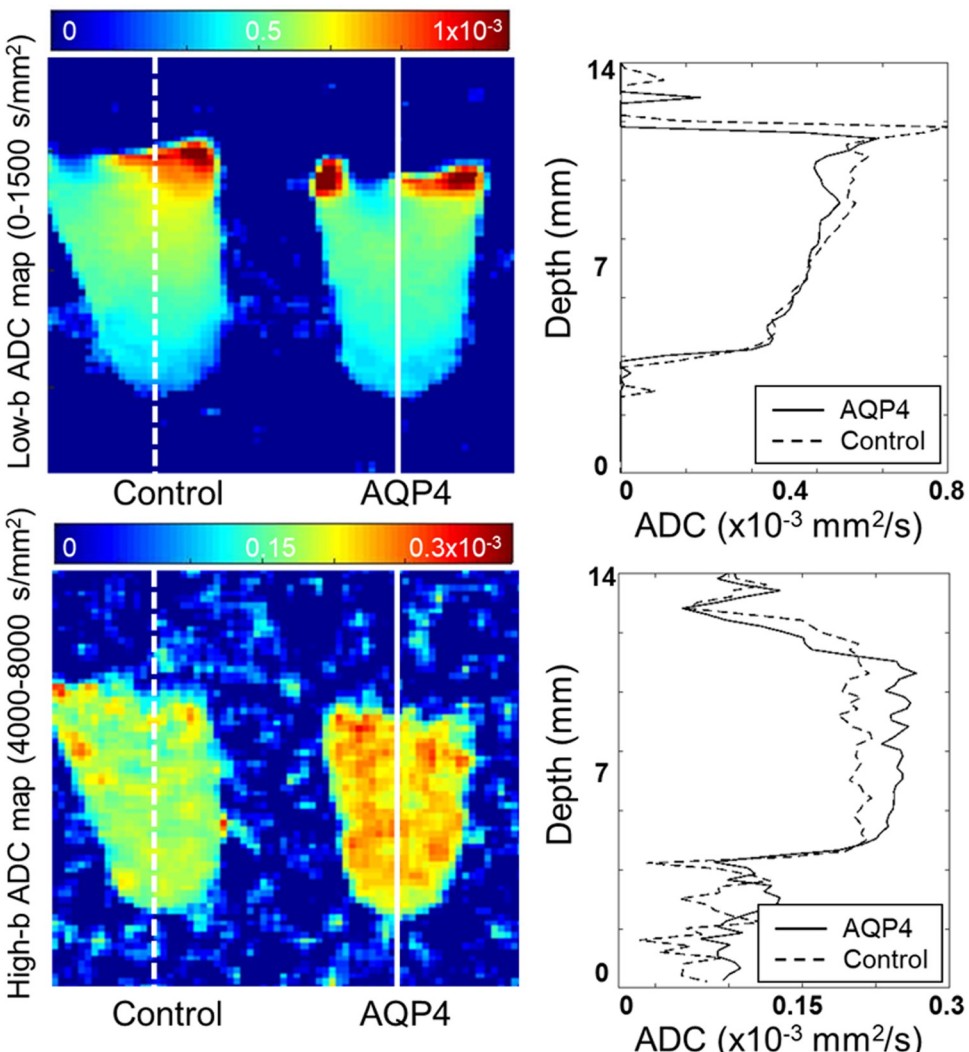

**Fig 3. Apparent diffusion coefficient (ADC) maps calculated for a low and high b-value ranges.** Apparent diffusion coefficient (ADC) maps of the samples are calculated for a low b-value range of 0–1500 s/mm$^2$ and a high b-value range of 4000–8000 s/mm$^2$. The low-b ADC map appears to depend on depth within the tube, while the high-b ADC map may be more sensitive to aquaporin-4 (AQP4) expression. Profiles along lines drawn on the ADC-maps (x $10^{-3}$mm$^2$/s) of aquaporin-4-expressing (AQP4, solid line) and -non-expressing cells (Control, dashed line) in PCR tubes are shown on the right.

both DKm model parameters K and D. On the other hand, K and D both varied with the depth (P<0.0001). There was also an interaction between the depth and cell type for K (P = 0.029). These results suggest that there are problems in independence and specificity when describing the data with this model.

This study was performed to test the suitability of two models in describing multi-b-value DWI of monoclonal cells. Although there are many DWI analysis models that could be applied to the data, as described by Novikov et al. [19], they are often just "representations" that fit the b-value-dependent signal change well but do not relate to the biology. Both models applied in this study have been widely used, but there have been many arguments about how they link with the biology. The DKm model is derived from a mathematical approximation to the signal without any biological information inserted, so it is reasonable that the parameter estimates do

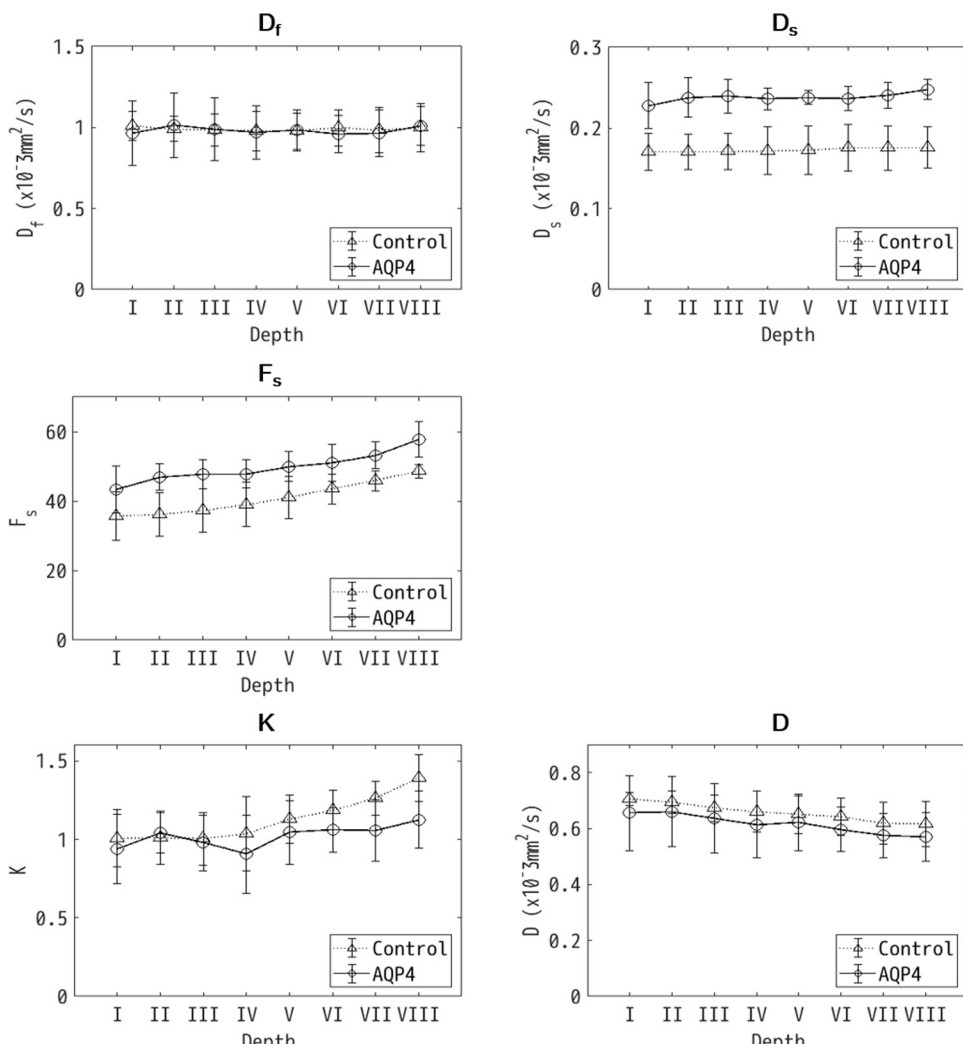

**Fig 4. Mean diffusion-weighted imaging (DWI) parameter estimates with standard deviations plotted against depth.** Mean DWI parameter estimates ($D_f$, diffusion coefficient of the fast compartment; $D_s$, diffusion coefficient of the slow compartment; $F_s$, volume fraction of the slow compartment; K, diffusion kurtosis; D, corrected diffusion coefficient) with standard deviations are plotted against depth (i.e. ROI number). The solid and dotted lines correspond to the aquaporin-expressing (AQP4) and -non-expressing (Control) cells, respectively. There is a significant difference between the AQP4 and Control samples for $D_s$ (P = 0.0003) and $F_s$ (P = 0.005). Also, $F_s$, K and D have a significant dependence on depth (P<0.0001). A significant interaction between the cell type and the depth was observed for K (P = 0.029).

**Table 1. Analysis-of covariance (ANCOVA) results (F-values and P-values) for the DWI parameter estimates.**

|  | 2-compartment model | | | Diffusion kurtosis imaging model | |
| --- | --- | --- | --- | --- | --- |
|  | $D_f$ | $D_s$ | $F_s$ | K | D |
| Cell type effect | 0.014(0.908) | 29.6(0.0003)* | 12.8(0.005)* | 1.62(0.232) | 0.763(0.403) |
| Depth effect | 0.420(0.886) | 1.59(0.152) | 27.9(<0.0001)* | 11.3(<0.0001)* | 10.1(<0.0001)* |
| Cell type x Depth | 0.438(0.875) | 0.667(0.699) | 0.558(0.787) | 2.40(0.029)* | 0.138(0.995) |

P-values are shown in parenthesis. Values less than 0.05 are considered significant (*). $D_f$, diffusion coefficient of the fast compartment; $D_s$, diffusion coefficient of the slow compartment; $F_s$, volume fraction of the slow compartment; K, diffusion kurtosis; D, corrected diffusion coefficient.

not correlate well with the cell type and cell density. On the other hand, based on the hypothesis that there are two major *in vivo* water compartments and diffusion in each compartment is Gaussian for the b-value range used in this study, there seems to be a link with the biology for the 2Comp model. However, there are a number of problems that might arise when performing bi-exponential curve fitting, so it should be used with extreme caution [19]. Although the samples used for this study were monoclonal cells, which are biologically much simpler than *in vivo* tissue, it is interesting that the parameter estimates of the 2Comp model were clearly linked to aspects of the biology (i.e. cell type and cell density). Although care should be taken when applying this model to complex *in vivo* structures, it might be useful for evaluating the state of tissues with a relatively simple structure.

## Limitations

There were some limitations in this study. First, AQP4 expression was not measured quantitatively. The AQP4 expression level can be measured with immunohistochemistry. It would be useful if a precise correlation between the expression level of AQP4 and one of the DWI parameters could be determined. A second limitation is that this study was an *in vitro* experiment. For *in vivo* DWI, the effect of perfusion at low b-values cannot be ignored. Further *in vivo* studies will be needed to clarify the relationship between the DWI parameters and the cell membrane water permeability, as well as the possible effects of perfusion.

## Conclusions

Differences in AQP4 expression affected DWI of cell suspensions. The 2Comp model was the more suitable model for our experiments on monoclonal cells. $D_s$ might be an effective biomarker for quantitatively evaluating cell membrane water permeability.

## Author Contributions

**Conceptualization:** Akiko Imaizumi, Takayuki Obata, Yasuhiko Tachibana.

**Data curation:** Akiko Imaizumi, Sayaka Shibata, Nobuhiro Nitta, Ichio Aoki.

**Formal analysis:** Akiko Imaizumi, Takayuki Obata.

**Funding acquisition:** Takayuki Obata.

**Investigation:** Akiko Imaizumi.

**Methodology:** Takayuki Obata, Yasuhiko Tachibana.

**Project administration:** Tatsuya Higashi.

**Resources:** Yoichiro Abe, Masato Yasui.

**Software:** Yasuhiko Tachibana.

**Supervision:** Yoichiro Abe, Ichio Aoki, Masato Yasui, Tatsuya Higashi.

**Validation:** Jeff Kershaw.

**Writing – original draft:** Akiko Imaizumi.

**Writing – review & editing:** Takayuki Obata, Jeff Kershaw.

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
