## [Decision Letter · Decision Letter 0]

26 Jan 2022

PONE-D-21-38062Quantitative measurement of diffusion-weighted imaging signal using expression-controlled aquaporin-4 cells: Comparative study of 2-compartment and diffusion kurtosis imaging modelsPLOS ONE

Dear Dr. Imaizumi,

Thank you for submitting your manuscript to PLOS ONE. After careful consideration, we feel that it has merit but does not fully meet PLOS ONE’s publication criteria as it currently stands. Therefore, we invite you to submit a revised version of the manuscript that addresses the points raised during the review process.

We look forward to receiving your revised manuscript.

Kind regards,

Kevin Camphausen

Academic Editor

PLOS ONE

Journal Requirements:

(This work was supported by grants from KAKENHI (15H04910, TO received.).

The funders had no role in study design, data collection and analysis, decision to publish, or preparation of the manuscript.)

Reviewers' comments:

Reviewer's Responses to Questions

**Comments to the Author**

1. Is the manuscript technically sound, and do the data support the conclusions?

Reviewer #1: Yes

Reviewer #2: Yes

2. Has the statistical analysis been performed appropriately and rigorously? 

Reviewer #1: Yes

Reviewer #2: Yes

3. Have the authors made all data underlying the findings in their manuscript fully available?

Reviewer #1: Yes

Reviewer #2: Yes

4. Is the manuscript presented in an intelligible fashion and written in standard English?

Reviewer #1: Yes

Reviewer #2: Yes

5. Review Comments to the Author

Reviewer #1: This manuscript is aimed at a very important aspect of DWI - understanding the relationship between biological tissue structure and the detected signal. The authors compared 2Comp and DKm models obtained from DWI of AQP4 expressing and non-expressing cells to look for biomarkers that indicate differences in cell membrane water permeability. The idea is straightforward and while the authors acknowledge two significant limitations (one being the fact that theAQP4 expression was not measured, which would have allowed for a more polished analysis with precise correlation between the expression level of AQP4 and the DWI parameters, the second being the in vitro aspect), the paper is nonetheless important to promote discussion and advance the field. To really emphasize the clinical implications of identifying biomarkers in this space, augmenting the intro to reflect the eventual clinical importance of this study, may be of value outside of the basic science context/audience.

Reviewer #2: The manuscript technically sound, and do the data support the conclusions.

The statistical analysis been performed appropriately and rigorously.

The authors made all data underlying the findings in their manuscript fully available.

The manuscript presented in an intelligible fashion and written in standard English.

6. PLOS authors have the option to publish the peer review history of their article (what does this mean?). If published, this will include your full peer review and any attached files.

Reviewer #1: No

Reviewer #2: No

---

## [Author Response · Author response to Decision Letter 0]

25 Feb 2022

Thank you very much for your helpful comments concerning our manuscript: PONE-D-21-38062.

According to the comments, the manuscript was revised as follows.

Journal Requirements

(JR-1-0)

Referring to the attached file, the manuscript body was fixed to "justified". (Page 2, Line 16, etc.)

(JR-1-1)

Referring to the PLOS ONE TITLE, AUTHOR, AFFILIATIONS FORMATTING GUIDELINES, title was fixed to “centered”. (Page 1, Line 1-3.)

(JR-1-2)

Referring to the PLOS ONE MANUSCRIPT BODY FORMATTING GUIDELINES, paragraphs were “indented”. (Page 2, Line 17, etc.)

(JR-1-3)

In Figs 2-4, figure titles were corrected to 15 characters or less, and figure legends were changed as well. (Fig 2; Page 9, Line 141-146. Fig 3; Page 9, Line 147 - Page 10, Line 153. Fig 4; Page 10, Line 158 – 166.)

(JR-1-4)

All equations were “centered”. (Page 7, Line 102 & 109.)

(JR-1-5)

The text in Table 1 was fixed to “justified” and “double-spaced”. (Page 11, Line 169.)

(JR-1-6)

The style of the “References” was corrected. (Page 16, Line 242 – Page 18, Line 297.)

(JR-2)

The corresponding author has the following ORCID iD: 0000-0002-2949-4687.

(This work was supported by grants from KAKENHI (15H04910, TO received.).

The funders had no role in study design, data collection and analysis, decision to publish, or preparation of the manuscript.)

(JR-3)

Our Funding Statement was amended according to your comment. One other grant was added (KAKENHI (20K08150, JK received.) and the Ministry of Education, Culture, Sports, Science and Technology (MEXT), Japanese government) to the cover letter.

(JR-4)

The style of the reference list was corrected. (Page 16, Line 242 – Page 18, Line 297.) There are no retracted references in the list. According to Reviewer #1’s comment, the introduction was revised and references were added [4, 6, 7, 9-11].

Reviewers' comments

5. Review Comments to the Author

Reviewer #1: This manuscript is aimed at a very important aspect of DWI - understanding the relationship between biological tissue structure and the detected signal. The authors compared 2Comp and DKm models obtained from DWI of AQP4 expressing and non-expressing cells to look for biomarkers that indicate differences in cell membrane water permeability. The idea is straightforward and while the authors acknowledge two significant limitations (one being the fact that theAQP4 expression was not measured, which would have allowed for a more polished analysis with precise correlation between the expression level of AQP4 and the DWI parameters, the second being the in vitro aspect), the paper is nonetheless important to promote discussion and advance the field. To really emphasize the clinical implications of identifying biomarkers in this space, augmenting the intro to reflect the eventual clinical importance of this study, may be of value outside of the basic science context/audience.

(CA-5-R1)

In response to the reviewer’s comment, the introduction has been revised to emphasize the clinical importance of this study. (Page3, Line 42 – 53.)

“Aquaporin channels are known to be associated with various diseases. For example, neuromyelitis optica (NMO) is an autoimmune disease targeting AQP4 [3]. AQP4 is also associated with brain tumors [4] and neurodegenerative diseases including Alzheimer's disease [5]. It is also known that changes to the expression of AQP4 can alter the accumulation of brain edema in ischemia [6, 7]. Other aquaporin subtypes are involved in various diseases such as tumors [8], cataracts [9], and nephrogenic diabetes insipidus [10, 11]. Unfortunately, a clinical imaging method that can evaluate aquaporin expression in vivo has not yet been established. Moreover, as there are many subtypes of aquaporin with diverse functions, it is probably more reasonable to evaluate the cell membrane water permeability rather than the expression of aquaporin itself. It is therefore expected that an imaging technique that can quantitatively evaluate changes in cell membrane water permeability will be useful in disease diagnosis.”

---

## [Editor Report · Decision Letter 1]

22 Mar 2022

Quantitative measurement of diffusion-weighted imaging signal using expression-controlled aquaporin-4 cells: Comparative study of 2-compartment and diffusion kurtosis imaging models

PONE-D-21-38062R1

Dear Dr. Imaizumi,

We’re pleased to inform you that your manuscript has been judged scientifically suitable for publication and will be formally accepted for publication once it meets all outstanding technical requirements.

Kind regards,

Kevin Camphausen

Academic Editor

PLOS ONE
---

## [Editor Report · Acceptance letter]

12 Apr 2022

PONE-D-21-38062R1 

Quantitative measurement of diffusion-weighted imaging signal using expression-controlled aquaporin-4 cells: Comparative study of 2-compartment and diffusion kurtosis imaging models 

Dear Dr. Imaizumi:

I'm pleased to inform you that your manuscript has been deemed suitable for publication in PLOS ONE. Congratulations! Your manuscript is now with our production department. 

Kind regards, 

on behalf of

Dr. Kevin Camphausen 

Academic Editor

PLOS ONE